# REFLECT: Summarizing Robot Experiences for FaiLure Explanation and CorrecTion

**Zeyi Liu**[*]        **Arpit Bahety**[*]        **Shuran Song**

Columbia University, New York, NY, United States

https://robot-reflect.github.io/

**Abstract:** The ability to detect and analyze failed executions automatically is crucial for an explainable and robust robotic system. Recently, Large Language Models (LLMs) have demonstrated strong reasoning abilities on textual inputs. To leverage the power of LLMs for robot failure explanation, we introduce REFLECT, a framework which queries LLM for failure reasoning based on a hierarchical summary of robot past experiences generated from multisensory observations. The failure explanation can further guide a language-based planner to correct the failure and complete the task. To systematically evaluate the framework, we create the RoboFail dataset with a variety of tasks and failure scenarios. We demonstrate that the LLM-based framework is able to generate informative failure explanations that assist successful correction planning.

**Keywords:** Large Language Model, Explainable AI, Task Planning

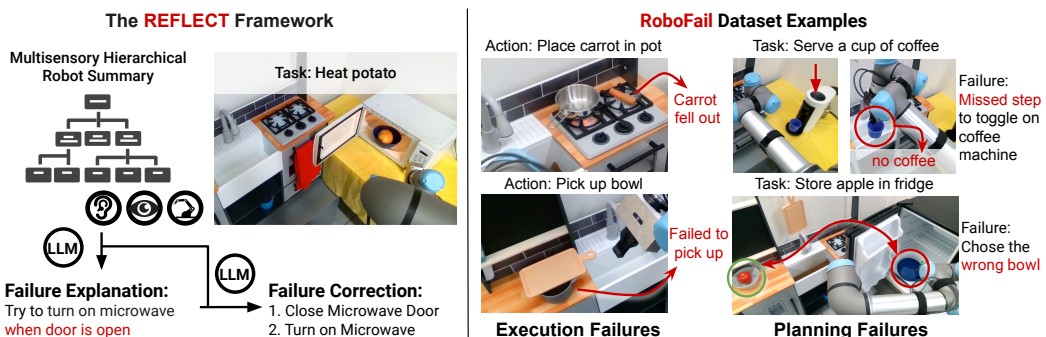

Fig 1: **A framework for robot failure explanation and correction.** On the left, we show the REFLECT framework that converts multisensory observations (RGB-D, audio, robot states) to a hierarchical summary of robot experiences. The summary is then used to query a Large Language Model (LLM) for failure explanation and correction. The right shows a few example failure cases in the RoboFail dataset.

## 1   Introduction

With the increasing expectations for robots to work on long-horizon tasks in complex environments, failures are inevitable. It is thus an essential capability for a robotic system to reflect on its past experiences and explain its failures in natural language. The failure explanations can either help a human user to debug the robotic system without having to read through the tedious execution logs, or guide the robot to correct the failure by itself.

We hypothesize that an effective failure reasoning framework requires several key components: first, a component to summarize "what happened" by converting *unstructured, multimodal* robot sensory data into a *structured, unified* format; second, a component to reason "what was wrong" by inferring from the summary whether expected outcomes of the robot plan were achieved; and finally, the ability to plan "what to do" based on the failure reasoning to correct the failure and complete the task.

---

[*]indicates equal contribution

7th Conference on Robot Learning (CoRL 2023), Atlanta, USA.

Recently, Large Language Models (LLMs) [1, 2, 3] have been shown to exhibit strong reasoning [4, 5, 6] and planning [7, 8, 9] capabilities, making it a promising component for explainable and robust robotic systems. However, the remaining challenges lie in how to generate a textual summary of robot sensory data and systematically query LLMs for failure reasoning. In other words, how do we transform the robot failure reasoning task into a language reasoning task? We observe two important attributes of a good robot summary:

- **Multisensory.** The summary should cover all sensory modalities the robot has access to, such as visual, audio, contact, etc. This is because certain failures can be more easily identified through one type of sensory data than another. For example, it is easier to determine if an object is dropped or if water is running from the faucet using auditory cues rather than visual ones.

- **Hierarchical.** To support effective failure explanations, the robot summary requires multiple levels of abstraction: to quickly localize the failure, the highest summary level should focus on identifying misalignment between the robot high-level plan and execution outcomes; while the lower summary levels should maintain enough environmental context for LLMs to generate an informative explanation that is useful for correction planning.

Based on these observations, we introduce **REFLECT**, a framework that summarizes robot experiences for failure explanation and correction. The framework first processes post-execution robot observations to generate a hierarchical summary with three levels of abstraction. Equipped with this summary, we propose a progressive failure explanation algorithm for failure reasoning. Through our experiments, we demonstrate that the framework is able to generate informative failure explanations as assessed by human evaluators, and also guide a language-based planner to generate correction plans for several failure scenarios.

To systematically evaluate REFLECT, we also create the **RoboFail** dataset, which includes 100 failure demonstrations generated in the AI2THOR simulation [10] and 30 real-world failure demonstrations collected with a UR5e robot arm. We hope the dataset will encourage development of more explainable and robust embodied AI systems.

## 2   Related Work

**Dense Video Captioning** for human activity videos has been a challenging task in computer vision. Recent video captioning methods typically train transformer-based models to jointly localize and caption events in videos [11, 12, 13, 14, 15]. Yet it remains a challenge to caption robot videos due to a lack of data. With the emergence of large foundation models, zero-shot video captioning is made possible [16, 17, 18]. These works combine VLMs and LLMs to caption egocentric human activity videos in a zero-shot manner. Extending upon prior works, our approach generates captions that are task-centric and action-centric for temporal robot sensory data in a zero-shot manner, which helps downstream tasks such as robot behavior analysis [19].

**Robot Failure Explanation** is an important task long studied in the HRI community to increase human trust in robotic systems [20, 21] and allow non-expert users to better assist robots under failure scenarios [22, 23]. However, prior works are limited as each only address a specific set of failure scenarios: Das and Chernova [23] specifically study failure cases in picking, Diehl and Ramirez-Amaro [24] study two pick-and-place tasks, Song et al. [25] focus on object navigation failures whereas Inceoglu et al. [26] focus on detecting failures in a few short-horizon object manipulation tasks. By leveraging the advanced reasoning ability of LLM, our framework is able to detect and explain a wide range of failure scenarios without assumptions on the task configuration or failure type.

**Task Planning with Large Language Models** Large Language Models can be leveraged to decompose high-level, abstract task instructions into low-level step-by-step actions executable by agents [7, 9, 27, 28]. Recent works have also demonstrated the self-reflective and self-corrective ability of LLMs based on environment feedback [8, 29, 30, 31, 32]. However, they all assume ground truth environment feedback associated with one or few actions. In this work, we explore the reflective ability of LLMs directly on multisensory robot observations and its temporal reasoning ability on long-horizon robot task executions. Our framework is able to directly operate on real-world robot task execution data with various failure scenarios.

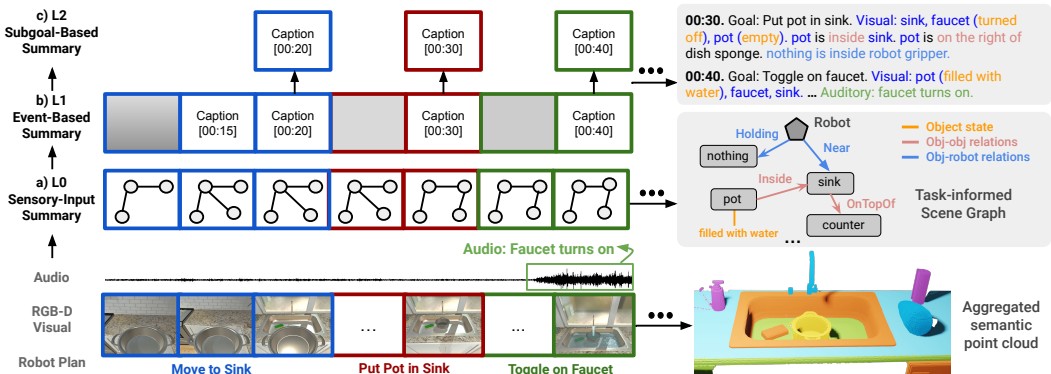

Fig 2: **Hierarchical robot summary** is composed of: a) a sensory-input summary that converts multisensory robot observations (RGB-D, sound, robot states) into task-informed scene graphs and audio summary; b) an event-based summary that generates captions for key event frames; c) a subgoal-based summary that contains the end frame of each subgoal.

# 3   Method: the REFLECT Framework

The REFLECT framework summarizes a robot's past experiences for failure explanation and correction. It contains three modules: a hierarchical robot summary module that summarizes multisensory robot data with three levels of abstraction (§3.1), a progressive failure explanation module that queries LLM to detect and explain the failure (§3.2), and finally a failure correction planner that generates an executable correction plan (§3.3).

## 3.1   Hierarchical Robot Summary

To perform effective failure explanation and correction, we propose a hierarchical summary structure to 1) aggregate and convert robot sensory data over time into a unified structure; 2) summarize the robot experiences for efficient failure localization and explanation. The hierarchical summary structure contains three levels: sensory-input summary, event-based summary, and subgoal-based summary.

### 3.1.1   Sensory-Input Summary

The sensory-input summary processes unstructured, multimodal robot sensory observations into a unified structure that stores necessary information for failure explanation.

**Visual summary with task-informed scene graphs.**   To understand a robot's interactions with its surrounding environment, it is important to extract inter-object relations, robot-object relations, and object state information from the observations. Given the RGB-D observation at timestep $t$, we run object detection on the RGB image to obtain a semantic segmentation $I_t^S$ and project it to a 3D semantic point cloud using the observed depth. In addition, for objects that can change states (e.g. microwave can be turned on and off), we crop the image based on the object's detected bounding box and compute the cosine similarity of the CLIP embedding [33] between the cropped image and a pre-defined list of object state labels (see details in §B.3). Given the semantic point cloud, heuristics are applied to compute inter-object spatial relations for 8 commonly-used spatial relations[1]: inside, on top of, on the left, on the right, above, below, occluding, and near. We also infer the robot-object relation from gripper state and object detection results. To aggregate the 3D point cloud over time frames, we use a similar approach as Li et al. [34] to align the newly observed point cloud $p_t$ with the accumulated point cloud from all previous time steps $P_{t-1}$ using 4 heuristic operations: add, update, replace, and delete.[1]

Once having the aggregated point cloud, we construct a scene graph $G_t = \{N + \{n_{robot}\}, E\}$, which describes the object nodes ($N$) and their spatial relations ($E$). Each node is defined as $n_i = (c_i, s_i)$, where $c_i$ is the object class, $s_i$ is the object state if any. Each edge contains the spatial relation between the two objects. We add the robot as an additional node $n_{robot}$ and a special relation "inside robot gripper". To make the

---

[1]Details of all heuristics can be found in appendix.

summary succinct and reduce computation and memory cost, we only consider objects that are relevant to the task (as extracted from the original robot plan), and objects spatially related to the task-relevant objects.

**Audio summary.** Audio is a useful modality for failure reasoning as it provides immediate feedback of failure events (e.g. something drops from gripper to the ground) and detects state changes when visual cues are limited (e.g. stove burner turns on but occluded by an object on top).

Given an input audio stream, we first segment the whole audio clip into several sub-clips by filtering out ranges where the volume is below a certain threshold $\epsilon$. Then for each sub-clip $s$, we compute its audio-lingual embedding with a pre-trained audio-language model (e.g. AudioCLIP [35], Wav2CLIP [36]). We calculate the cosine similarity between the audio embedding and the CLIP embeddings for a list of candidate audio event labels $L$, where the highest-scoring label $l^*$ is selected: $l^* = \text{argmax}_{l \in L}[C(s,l)]$, $C = \frac{f_1(s) \cdot f_2(l)}{||f_1(s)||||f_2(l)||}$, where $f_1$ is the embedding function for audio, $f_2$ is the embedding function for text.

### 3.1.2 Event-Based Summary

Given that the sensory-input summary (§3.1.1) computes a scene graph for each frame and thus contains redundant information, the goal of the event-based summary is to select key frames and generate text captions from the corresponding scene graphs.

We design a key frame selection mechanism based on visual, audio, and robot states. More specifically, a frame is selected if it satisfies any of the conditions below: 1) The task-informed scene graph of the current frame $G_t$ is different from the previous frame $G_{t-1}$. 2) The frame is the start or end of an audio event. 3) The frame marks the end of a subgoal execution.

For each key frame, we convert the scene graph into text with the following format.[2] When constructing the visual observation, we only consider objects that are visible in the current frame.

> [timestep] Action: [robot action]
> Visual observation: object1 [state], object2, object3 [state] ... # objects and states
> object1 is [spatial relation] object2 ... # inter-object relations
> object3 is inside robot gripper. # robot-object relations
> Auditory observation: [audio summary].

### 3.1.3 Subgoal-Based Summary

The event-based summary (in §3.1.2) stores environment observations throughout the robot task execution. However, it's hard for LLM to infer the expected outcomes and identify failures for every one of the low-level actions. As a result, we introduce the subgoal-based summary, which consists of observations at the end of each subgoal, for LLM to identify misalignment between the robot execution outcomes and its high-level plan (e.g. move to the toaster, put bread slice in the toaster). The subgoal-based summary enables the failure explanation module to quickly process the robot experience summary by checking whether each subgoal is satisfied while ignoring low-level execution details. Once a failure is detected, relevant environment information stored in the event-based summary or sensory-input summary can be retrieved for detailed failure explanation.

### 3.2 Progressive Failure Explanation

The failure explanation algorithm should handle both execution and planning failures, where the former requires action-level observation details and the latter requires task-level information such as task description and robot plan. To do so, the algorithm first identifies the type of failure and then retrieves relevant information from the hierarchical summary to construct the query to LLM. As shown in Fig. 3, the algorithm first iterates through the subgoals and verifies success using the following prompt: [3]

> The robot subgoal is [robot subgoal at time $t$]. Given [subgoal-based summary at time $t$]
> Q: Is the subgoal satisfied? A: Yes

---

[2] text color: blue: visual , green: audio , light blue: contact, yellow: summary, orange: final state, failure explanation, brown: timestep, task name, robot subgoal, original robot plan, goal state, blue highlight: LLM output

[3] Examples of full prompts are shown in the appendix.

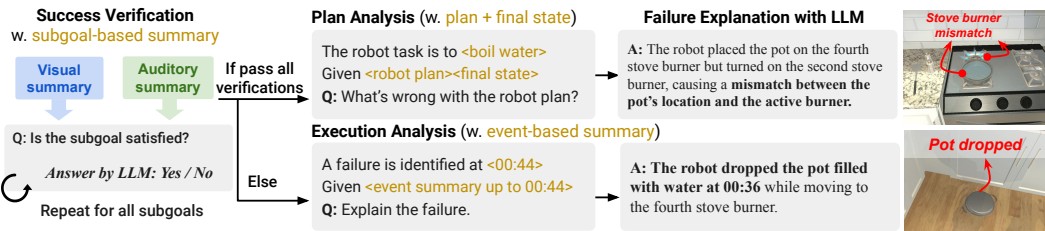

Fig 3: **Progressive failure explanation** verifies success for each subgoal. If a subgoal fails, the algorithm enters the *execution analysis* stage for detailed explanation. If all subgoals are satisfied, the algorithm enters *planning analysis* stage to check errors in the robot plan.

The LLM is instructed to output 'Yes' or 'No'. If a subgoal is not achieved, then we retrieve history observations stored in event-based summary for failure explanation as follows:

> The robot task is to [task name]. A failure is identified at $t$. Given [event-based summary up to $t$]
> Q: Briefly explain what happened at $t$ and what caused the failure?
> A: At 00:44, the robot attempted to put the pot on the fourth stove burner, but the pot was not in its gripper. The failure was caused by the robot dropping the pot filled with water at 00:36 while moving to the fourth stove burner.

In case all subgoals are achieved but the task still failed, then it's likely that the plan itself was incorrect. We use the robot original plan and the final state of the environment to identify errors in the robot plan. The final state is obtained from the scene graph generated from the aggregated semantic point cloud in the last time step without view-specific relations (on the left, on the right, occluding).

> The robot task is to [task name]. The task is successful if [goal state].
> The robot plan is [original robot plan]. Given [final state]
> Q: What's wrong with the robot plan that caused the robot to fail?
> A: The robot placed the pot on the fourth stove burner but turned on the second stove burner, causing a mismatch between the pot's location and the active burner.
> Q: Which time step is most relevant to the above failure?
> A: 00:49

### 3.3 Failure Correction Planner

A failure correction planner should generate an executable plan for the robot to correct the failure and complete the task, starting from the final state of the original task execution. Prior work [22] has shown that good failure explanations help non-expert users better understand the failure and assist the robot. Analogously, we hypothesize that the failure explanation can also guide a language planner to generate a high-level correction plan that leads to task success. The prompt is formatted as below:[3]

> The robot task is to [task name]. The original robot plan is [original robot plan].
> Given [failure explanation] [final state] and [goal state]
> Correction plan: toggle_off (stoveburner-2), toggle_on (stoveburner-4)

To make sure the plan generated by the language model is executable in the environment, we adopt the idea of Huang et al. [28] to map each LLM-generated action to its closest executable action in the task environment using a large pre-trained sentence embedding model.

## 4 The RoboFail Dataset

In simulation, we generate task execution data in AI2THOR and manually inject failures. The dataset contains a total of 100 failure scenarios, with 10 cases for each of the 10 tasks (see details in §B.1). We store the RGB-D observations, sound (20 classes in total), robot state data, as well as ground truth metadata obtained from simulation. The real-world dataset is collected by human teleoperation of a UR5e robot arm in a toy kitchen environment. The dataset contains 11 tasks with a total of 30 failure scenarios. We store the RGB-D observations (with Intel RealSense D415), recorded sound (with RØDE VideoMic Pro+), and robot proprioception data. A taxonomy of failure scenarios are visualized in Fig. 4.

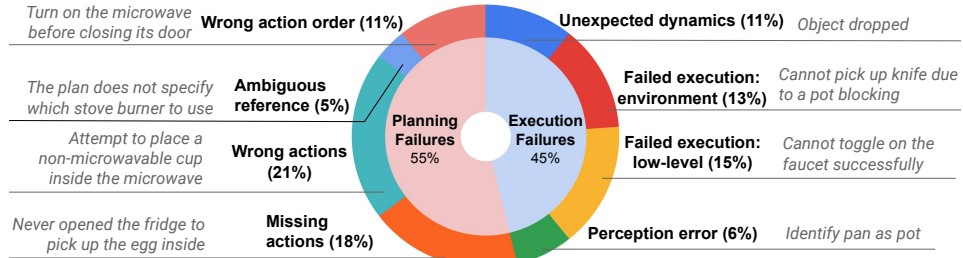

Fig 4: **RoboFail Failure Taxonomy**

## 5 Evaluation

We systematically evaluate the ability of REFLECT to localize, explain and correct robot failures. In AI2THOR simulation, the agent interacts with the environment through action primitives, such as pick up, toggle on, move left. We assume the framework has access to ground truth object detection and state detection in simulation. We also evaluate the ability of the framework to summarize real-world robot sensory data. The real-world failure data is collected by a human teleoperating a UR5e robot arm to mimic robot policies according to a provided high-level plan. We use MDETR [37] for object detection, CLIP [33] for object state detection, and AudioCLIP [35] for sound detection. We use GPT-4 [38] as the LLM. The below metrics are evaluated in our experiments:

- Exp (explanation): percentage of predicted failure explanations that are correct and informative as determined by human evaluators[4].
- Loc (localization): percentage of predicted failure time that align with actual failure time. A predicted time is considered aligned if it falls within the annotated failure time ranges in the dataset.
- Co-plan (correction planning success rate): percentage of tasks that succeed after executing the correction plan. Task success is determined by comparing the final state and the specified goal condition.

To demonstrate advantages of our framework, we compare with the following baselines/ablations:

- BLIP2 caption: caption key frames with BLIP2, a state-of-the-art image caption model [39].
- w/o sound: our approach without the audio modality.
- w/o progressive: similar to open-ended Q&A in Socratic Models [16], directly query LLM for failure explanation given the robot summary without progressive failure explanation.
- Subgoal only: using only subgoal-based summary for failure explanation.
- LLM summary: convert all sensory-input summary to text and prompt LLM to summarize the text for failure explanation.
- w/o explanation: query LLM for a correction plan without failure explanation.

By evaluating REFLECT on the RoboFail dataset and comparing with the above baselines/ablations, we have the below findings:

**REFLECT is able to generate informative failure explanations that assist correction planning.** Tab. 1 and 2 summarize the evaluation result, where RE-FLECT achieves the highest performance in explaining, localizing, and correcting failures in both simulation and the real world. The performance slightly decreases in real world due to perception errors. We find that localization is slightly harder for planning failures as the failure is usually not associated with a single time step. In simulation, our framework achieves around 80% correction planning success rates for both failure types.

| Method | Execution failure | | | Planning failure | | |
|---|---|---|---|---|---|---|
| | Exp | Loc | Co-plan | Exp | Loc | Co-plan |
| w/o progressive | 46.5 | 62.8 | 60.5 | 61.4 | 70.2 | 64.9 |
| Subgoal only | 76.7 | 74.4 | 51.2 | 71.9 | 73.7 | 75.4 |
| LLM summary | 55.8 | 67.4 | 65.1 | 57.9 | 54.4 | 66.7 |
| w/o explanation | - | - | 41.9 | - | - | 56.1 |
| REFLECT | **88.4** | **96.0** | **79.1** | **84.2** | **80.7** | **80.7** |

Table 1: Result in Simulation Environments

**Audio data is useful for failure explanation.** As shown in Tab. 2, the explanation and localization accuracy for [w/o sound] both decrease around 20% on execution failures as compared to REFLECT. This is because some unexpected events (e.g. object dropping on floor) or object states hard to identify using visual detectors (e.g. stove burner occluded by a pot on top) are easier to be detected through audio.

---

[4]Details can be found in appendix.

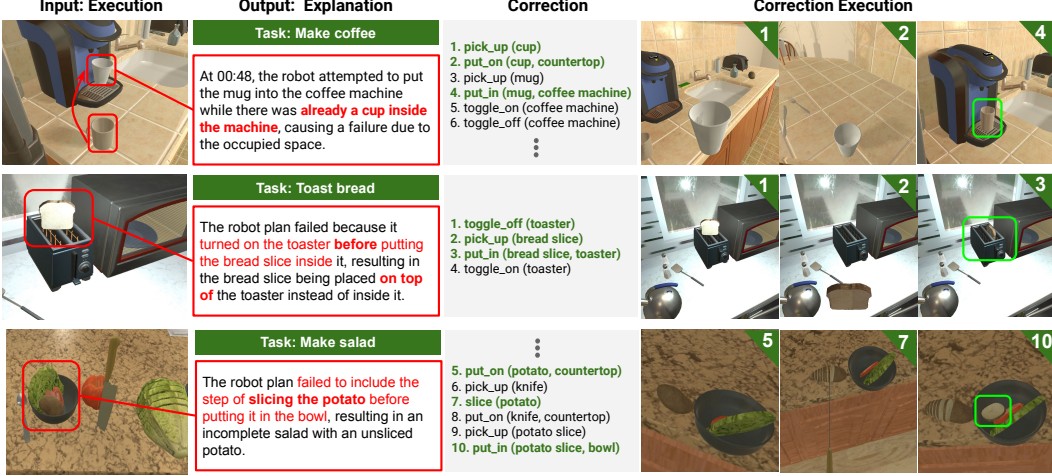

Fig 5: **Qualitative results in simulation.** Given a failed robot task execution, REFLECT is able to generate informative failure explanations for both execution and planning failures. Conditioned on the explanation, a language planner can generate a high-level plan for the robot to correct the failure and complete the task.

**Task-relevant object spatial and state information is crucial.** As shown in Tab. 2, [BLIP2 caption] achieves the worst performance in all scenarios because the captions generated by BLIP2 lack necessary information for failure explanation. In contrast, our zero-shot caption method is designed to capture environment information such as object states and spatial relations, which are task-relevant and crucial for failure explanation. The figure on the right shows that REFLECT is able to summarize object states such as "fridge (with door open)" and spatial relations such as "apple inside white bowl", which are not present in the BLIP2 caption.

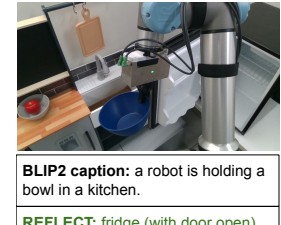

**BLIP2 caption:** a robot is holding a bowl in a kitchen.

**REFLECT:** fridge (with door open), apple, white bowl, dark blue bowl. apple is inside white bowl. dark blue bowl is inside robot gripper.

**Progressive failure explanation is important.** Our progressive failure explanation algorithm leverages the hierarchical summary to first identify the failure with the subgoal-based summary, and then query LLM for failure explanation accordingly. Comparing to [w/o progressive] in Tab. 1 and 2, the progressive algorithm helps with more accurate localization and informative explanation. To better understand the difference, a qualitative example is shown in Fig. 6: although [w/o progressive] mentions

| Method | Execution failure | | Planning failure | |
|---|---|---|---|---|
| | Exp | Loc | Exp | Loc |
| BLIP2 caption | 6.25 | 25.0 | 35.7 | 57.1 |
| w/o sound | 50.0 | 68.8 | 78.6 | 78.6 |
| w/o progressive | 43.8 | 81.3 | 71.4 | 78.6 |
| Subgoal only | 56.3 | 62.5 | 71.4 | 78.6 |
| LLM summary | 37.5 | 75.0 | 64.3 | 71.4 |
| REFLECT | **68.8** | **93.8** | **78.6** | **78.6** |

Table 2: Result in Real-world Environments

that the task failed because the robot did not have an egg in its gripper to put in the pan, it does not reason why the egg was not present. In contrast, REFLECT identifies failure in the "pick up egg" action and then queries the event-based summary to infer the exact failure cause – "fridge was closed".

**Hierarchical structure is important.** The performance decrease of [Subgoal only] shows the importance of event-based summary as it stores intermediate environment observations that are useful for failure explanation. Consider the scenario when the robot accidentally dropped the pot it was holding when moving to stove burner. [Subgoal only] only infers that the object was not in the robot gripper at the end of the subgoal execution. In contrast, the event-based summary stores auditory observation of "something drops on ground" and visual observation of "nothing is inside robot gripper", which helps identify that the pot was dropped and the exact time step the failure occurred.

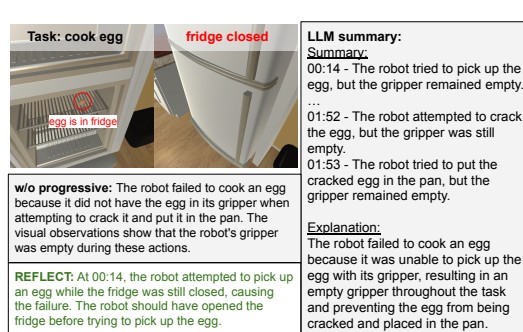

Fig 6: [w/o progressive] vs. [LLM summary] vs. Ours

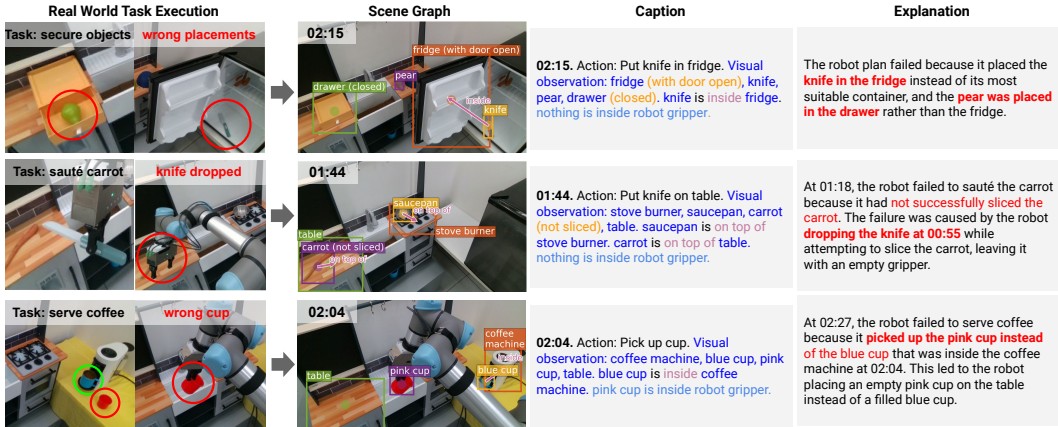

Fig 7: **Qualitative results in real world.** REFLECT is able to summarize and generate informative failure explanations for real-world robot executions. The above figure shows three failed task executions on the left, the corresponding scene graph and caption for one key frame in the middle, and the LLM-generated failure explanation on the right.

In addition, the hierarchical structure is a better way to condense the sensory-input summary for failure explanation. We implement an alternative [LLM summary], which prompts LLM to summarize the sensory-input summary. The performance decreases significantly in both simulation and real world as the LLM-generated summary loses information that is relevant for failure explanation. As shown in Fig. 6, the summary and explanation of [LLM summary] only mention that the robot tried to pick up the egg and the gripper remained empty, but did not mention that the fridge was closed, which is the actual failure cause.

**Failure explanation helps correction.** From Tab. 1, we observe that the correction planning success rate significantly decreases for [w/o explanation]. This is because the failure explanation can guide LLM to generate a correction plan based on the failure cause. As shown in Fig. 8, given the failure reason that the mug cannot be put inside the coffee machine due to presence of a cup, RE-FLECT generates a plan to move the cup away and then proceed with the task. Whereas [w/o explanation] simply repeats the original plan without taking any actions to address the cause of failure.

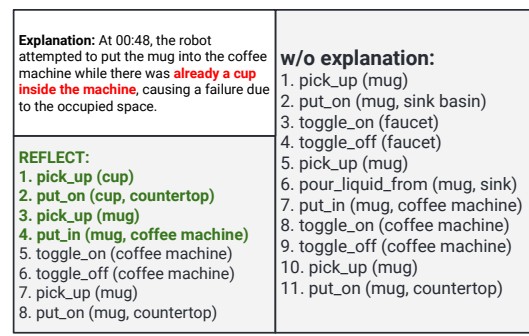

Fig 8: Failure explanation helps correction planning.

**Limitations.** There are a few limitations in the method used to convert sensory data into textual summary. Even though the heuristics used to generate scene graphs is sufficient for scenarios studied in the paper, it may fall short in more complex environments. Either training a large spatial reasoning model or fine-tuning an existing model on robotics data could be a promising solution [40, 41, 42]. In addition, the object state detection method assumes a given list of candidate object states, which can be potentially relaxed by a method (e.g. prompting a LLM) that output possible states given the object category.

The framework also assumes the rest of the environment will remain static throughout the robot task execution. Finally, given the information (object detection, object states, spatial relations) that the robot summary contains, it is less effective for handling low-level control failures. Future work may consider developing better perception methods that capture more low-level state information.

# 6   Conclusion

We propose a framework, REFLECT, which converts multisensory observations into a hierarchical summary of robot past experiences and queries LLM progressively for failure explanation. The generated explanation can then guide a language planner to correct the failure and complete the task. To evaluate the framework, we create a dataset of robot failed executions in both simulation and real world and show that REFLECT achieves better performance as compared to several baselines and ablations. We encourage future work to extend upon the framework and explore more use cases of the robot summary.

**Acknowledgments**

We would like to thank Cheng Chi and Zhenjia Xu for their help in setting up real world experiments, and Huy Ha, Mandi Zhao, Samir Gadre, Mengda Xu, Dominik Bauer for valuable discussions and feedback. This work was supported in part by NSF Award #2143601, #2037101, and #2132519. We would like to thank Google for the UR5 robot hardware. The views and conclusions contained herein are those of the authors and should not be interpreted as necessarily representing the official policies, either expressed or implied, of the sponsors.

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

# Appendix

## A Method Details

### A.1 Spatial Relation Heuristics

We implement a total of 8 object spatial relation heuristics given the aggregated semantic point cloud and 3D object bounding boxes. We consider two in-contact relations when the minimum distance between the object point clouds is smaller than 5cm:

1. **Inside.** Object A is considered inside object B if over 50% percent of object A's point cloud is inside the convex hull of object B.

2. **On top of.** Object A is considered on top of object B if it satisfies the following two conditions: a) over 70% of object A's point cloud lies within a XY-plane projected from object B's 3D bounding box. b) over 70% of object A's points are above the upper Z bound of object B's bounding box.

If the object point cloud distance is larger than 5cm but smaller than 40cm, we subtract the center points of object A and B's 3D bounding box, transform it back to camera coordinates (Y-up) and normalize to get a 3 dimensional unit vector. The following relations are considered:

1. **Above & Below.** If the y-component of the vector is over 0.9, then object A is above object B. If the y-component is smaller than -0.9, then object A is below object B.

2. **On the left & On the right.** If the x-component of the vector is over 0.8, then object A is on the right of object B. If the x-component is smaller than -0.8, then object A is on the left of object B.

3. **Occluding.** If 90% of object A's points have a smaller depth than object B's minimum depth and object A's projected 2D bounding box (projected in current image space from its aggregated 3D point cloud) overlaps more than 50% with that of object B's projected bounding box, then object B is considered occluding object A.

4. **Near.** If none of the above relations satisfy but the object distance is smaller than 10cm, then object A is near object B.

### A.2 Scene Graph Aggregation Heuristics

We aggregate the 3D point cloud over time frames with a similar approach as Li et al. [34]. Consider the point cloud observed in the current time step $p_t$ and the accumulated point cloud from all previous time steps $P_{t-1}$, we can obtain the accumulated point cloud $P_t$ up to time step $t$ with 4 operations: ADD, UPDATE, REPLACE, and DELETE.

1. **ADD.** If an object is observed in $p_t$ but not in $P_{t-1}$, we consider it a newly appeared object and add it as a new node. If the node is task-relevant, then its relations with all existing objects will be computed and added as edges. If the node is not task-relevant, then its relations with existing task-relevant objects will be computed and added as edges.

2. **UPDATE.** If an object is in both $p_t$ and $P_{t-1}$, and the object point cloud is aligned, we update the object point cloud by concatenating the newly observed points and updating its existing edges with other objects by recomputing the spatial relations.

3. **REPLACE.** If an object is in both $p_t$ and $P_{t-1}$, but the misalignment of the object point clouds is larger than a threshold $d$, indicating the object has moved. We remove the old node in $P_{t-1}$ and add the object as a new node.

4. **DELETE.** If an object is in $P_{t-1}$ but not in $p_t$, and the robot is interacting with the object, then we remove the object from the accumulated point cloud since its location becomes unknown.

### A.3 Example Prompts

### A.3.1 Subgoal Verification

system prompt:

> You are a success verifier that outputs 'Yes' or 'No' to indicate whether the robot goal is satisfied given the robot observations.

user prompt:

> The robot goal is to [SUBGOAL]. Here are the robot observations after execution: [OBSERVATION]
> Q: Is the goal satisfied?
> A: Yes

The [SUBGOAL] and [OBSERVATION] entry will be filled in when prompting for each subgoal. Here's a complete example of a failed execution during the task 'boil water'. The robot accidentally dropped the pot when navigating to the stove burner and then attempted to put the pot on the stove burner:

> The robot goal is to pick up pot. Here are the robot observations after execution:
> Visual observation: pot (empty and clean). pot is inside robot gripper.
> Q: Is the goal satisfied?
> A: Yes

> The robot goal is to put pot in sink. Here are the robot observations after execution:
> Visual observation: faucet (turned off), pot (empty and clean), sink. pot (empty and clean) is inside sink. pot (empty and clean) is on the right of soap bottle. nothing is inside robot gripper.
> Q: Is the goal satisfied?
> A: Yes

> The robot goal is to toggle on faucet. Here are the robot observations after execution:
> Visual observation: pot (filled with water and clean), faucet (turned on), sink. pot (filled with water and clean) is inside sink. pot (filled with water and clean) is on the right of soap bottle. nothing is inside robot gripper. Auditory observation: water runs in sink.
> Q: Is the goal satisfied?
> A: Yes

> The robot goal is to toggle off faucet. Here are the robot observations after execution:
> Visual observation: pot (filled with water and clean), faucet (turned off), sink. pot (filled with water and clean) is inside sink. pot (filled with water and clean) is on the right of soap bottle. nothing is inside robot gripper.
> Q: Is the goal satisfied?
> A: Yes

> The robot goal is to pick up pot. Here are the robot observations after execution:
> Visual observation: pot (filled with water and clean), faucet (turned off), sink. pot is inside robot gripper.
> Q: Is the goal satisfied?
> A: Yes

> The robot goal is to put pot on fourth stove burner. Here are the robot observations after execution:
> Visual observation: second stove burner (turned off), first stove burner (turned off), third stove burner (turned off), fourth stove burner (turned off). nothing is inside robot gripper.
> Q: Is the goal satisfied?
> A: No

Here a subgoal is not satisfied, the program enters the **execution analysis** mode: history observations stored in the event-based summary are retrieved to query LLM for failure explanation.

### A.3.2 Failure explanation: execution analysis

system prompt:

> You are expected to provide explanation for a robot failure. You are given the robot actions and observations so far. Briefly explain the failure in 1-2 sentence. Mention relevant time steps if possible.

user prompt:

> The robot task is to boil water. At 00:44, a failure was identified.
>
> [Robot actions and observations before 00:44]
> 00:01. Action: Move to pot. Visual observation: nothing is inside robot gripper.
> 00:11. Action: Move to pot. Visual observation: faucet (turned off). nothing is inside robot gripper.
> 00:15. Action: Move to pot. Visual observation: pot (empty and clean). pot (empty and clean) is on the left of potato. pot (empty and clean) is on top of third countertop. nothing is inside robot gripper.
> 00:18. Action: Pick up pot. Visual observation: pot (empty and clean). pot is inside robot gripper.
> 00:21. Action: Move to sink. Visual observation: pot (empty and clean), faucet (turned off), sink. pot is inside robot gripper.
> 00:22. Action: Move to sink. Visual observation: pot (empty and clean), faucet (turned off), sink. pot is inside robot gripper.
> 00:25. Action: Put pot in sink. Visual observation: pot (empty and clean), faucet (turned off), sink. pot (empty and clean) is inside sink. pot (empty and clean) is on the right of soap bottle. nothing is inside robot gripper.
> 00:28. Action: Toggle on faucet. Visual observation: pot (filled with water and clean), faucet (turned on), sink. pot (filled with water and clean) is inside sink. pot (filled with water and clean) is on the right of soap bottle. nothing is inside robot gripper. Auditory observation: water runs in sink.
> 00:31. Action: Toggle off faucet. Visual observation: pot (filled with water and clean), faucet (turned off), sink. pot (filled with water and clean) is inside sink. pot (filled with water and clean) is on the right of soap bottle. nothing is inside robot gripper.
> 00:34. Action: Pick up pot. Visual observation: pot (filled with water and clean), faucet (turned off), sink. pot is inside robot gripper.
> 00:36. Action: Move to fourth stove burner. Visual observation: pot (empty and clean), faucet (turned off), sink. pot (empty and clean) is on the right of potato. pot (empty and clean) is on top of third countertop. nothing is inside robot gripper. Auditory observation: something drops.
> 00:38. Action: Move to fourth stove burner. Visual observation: pot (empty and clean), faucet (turned off), sink. nothing is inside robot gripper.
> 00:42. Action: Move to fourth stove burner. Visual observation: second stove burner (turned off), fourth stove burner (turned off), third stove burner (turned off), first stove burner (turned off). nothing is inside robot gripper.
> 00:43. Action: Move to fourth stove burner. Visual observation: second stove burner (turned off), fourth stove burner (turned off), third stove burner (turned off), first stove burner (turned off). nothing is inside robot gripper.
>
> [Observation at the end of 00:44]
> Action: Put pot on fourth stove burner. Visual observation: second stove burner (turned off), fourth stove burner (turned off), third stove burner (turned off), first stove burner (turned off). nothing is inside robot gripper.
>
> Q: Infer from [Robot actions and observations before 00:44] or [Observation at the end of 00:44], briefly explain what happened at 00:44 and what caused the failure.
> A: At 00:44, the robot attempted to put the pot on the fourth stove burner, but the pot was not in its gripper. The failure was caused by the robot dropping the pot filled with water at 00:36 while moving to the fourth stove burner.

The failure steps 00:36 and 00:44 can be extracted from the answer by prompting LLM to extract time steps from the output failure explanation.

### A.3.3 Failure explanation: planning analysis

In case all subgoals are satisfied, then there's likely mistakes in the robot original plan. The program will enter **planning analysis** mode. Take the failure scenario when the robot plan is wrong during the task 'boil water' as the robot placed the pot on one stove burner but toggled on another:

system prompt:

You are expected to provide explanation for a robot failure. You are given the current robot state, the goal condition, and the robot plan. Briefly explain what was wrong with the robot plan in 1-2 sentence.

user prompt:

The robot task is to boil water. The task is considered successful if a pot is filled with water, the pot is on top of a stove burner that is turned on.
Here's the robot observation at the end of the task execution:
faucet (turned off), second stove burner (turned on), sink, pot (filled with water and clean), fourth stove burner (turned off), third stove burner (turned off), first stove burner (turned off). pot (filled with water and clean) is on top of fourth stove burner (turned off). nothing is inside robot gripper.
The robot plan is:
00:18. Goal: Pick up pot.
00:25. Goal: Put pot in sink.
00:28. Goal: Toggle on faucet.
00:31. Goal: Toggle off faucet.
00:34. Goal: Pick up pot.
00:46. Goal: Put pot on stove burner.
00:49. Goal: Toggle on stove burner.

Q: Known that all actions in the robot plan were executed successfully, what's wrong with the robot plan that caused the robot to fail?
A: The robot placed the pot on the fourth stove burner but turned on the second stove burner, causing a mismatch between the pot's location and the active burner.

The failure time step can be obtained by a follow-up query to the LLM with the prompt below:

Q: Which time step is most relevant to the above failure?
A: 00:49

### A.3.4 Correction

Still take the failure scenario when the robot plan is wrong so that the robot placed the pot on one stove burner but toggled on another. A complete prompt for generating the failure correction plan is as follows:

system prompt:

Provide a plan with the available actions for the robot to recover from the failure and finish the task.
Available actions: pick up, put in some container, put on some receptacle, open (e.g. fridge), close, toggle on (e.g. faucet), toggle off, slice object, crack object (e.g. egg), pour (liquid) from A to B. The robot can only hold one object in its gripper, in other words, if there's object in the robot gripper, it can no longer pick up another object.
The plan should 1) not contain any if statements 2) contain only the available actions 3) resemble the format of the initial plan.

user prompt:

Task: boil water
Initial plan:
1. pick_up (pot)
2. put_in (pot, sink)
3. toggle_on (faucet)
4. toggle_off (faucet)
5. pick_up (pot)
6. put_on (pot, stove burner)
7. toggle_on (stove burner)
Failure reason: The robot placed the pot on the fourth stove burner but turned on the second stove burner, causing a mismatch between the pot's location and the active burner.
Current state: sink, pot (filled with water and clean), fourth stove burner (turned off), third stove burner (turned off), faucet (turned off), first stove burner (turned off), second stove burner (turned on). pot (filled with water and clean) is on top of fourth stove burner (turned off). nothing is inside robot gripper.
Success state: a pot is filled with water, the pot is on top of a stove burner that is turned on.
Correction plan: toggle_off (stoveburner-2), toggle_on (stoveburner-4)

# B Evaluation Details

## B.1 Dataset Details

Descriptions for the 10 simulation tasks and 11 real-world tasks in the RoboFail dataset are shown below.

| Task | Task Description / Goal State |
|---|---|
| boil water | A pot is filled with water, the pot is on top of a stove burner that is turned on |
| toast bread | A bread slice is inside a toaster that is turned on |
| fry egg | A cracked egg is in a pan, the pan is on top a stove burner that is turned on |
| heat potato | A potato is on a plate and inside a microwave that is turned on |
| serve coffee | A clean mug is filled with coffee and on top of the countertop |
| store egg | A bowl with an egg is stored inside the fridge |
| make salad | A bowl of sliced lettuce, tomato and potato is stored inside the fridge |
| water plant | The house plant is filled with water |
| switch devices | Laptop is closed on the TV stand and the television is turned on |
| serve warm water | A mug of water is heated in the microwave and served on the dining table |

Table 3: Simulation Tasks

| Task | Task Description / Goal State |
|---|---|
| boil water | A pot is filled with water, the pot is on top of a stove burner that is turned on |
| sauté carrot | A sliced carrot is inside a pan, the pan is on top of a stove burner that is turned on |
| heat potato | A potato is heated in the microwave and then put on the countertop |
| serve coffee | A mug is filled with coffee and on top of the countertop |
| store egg | A bowl with an egg is stored inside the fridge |
| secure objects | Knife is stored in a drawer and pear is stored in the fridge |
| apple in bowl | Apple is inside bowl |
| pear in drawer | Pear is inside a closed drawer |
| cut carrot | Carrot is sliced |
| fruits in bowl | All visible fruits are inside a bowl |
| heat pot | A pot is on top of a stove burner that is turned on |

Table 4: Real-world Tasks

## B.2 Comparison with GPT-3.5

We use GPT-4 for evaluation in our experiments. Here we provide a comparison of performance with the more accessible GPT-3.5. We have observed that GPT-4 exceeds GPT-3.5 in terms of both failure reasoning and planning abilities. The failure localization accuracy decreases as GPT-3.5 is less capable than GPT-4 to process irrelevant information in the summary and thus often wrongly local-

| Method | Execution failure | | Planning failure | |
| | Loc | Co-plan | Loc | Co-plan |
|---|---|---|---|---|
| GPT-3.5 | 47.8 | 30.4 | 51.9 | 40.7 |
| GPT-4 | **68.8** | **93.8** | **78.6** | **78.6** |

Table 5: Comparison with GPT-3.5

ize the failure. The correction planning success rate with GPT-3.5 decreases more significantly due to the combined factors of less accurate failure explanations and less planning ability even given correct failure explanations. We also show one qualitative example of the failure explanations generated by GPT-4 and GPT-3.5, which shows that GPT-4 is better at reasoning the root cause of the failure than GPT-3.5:

**GPT-3.5**: The robot plan failed because the robot did not successfully put the bread slice in the toaster, even though it successfully turned on the toaster.
**GPT-4**: The robot plan failed because it turned on the toaster before putting the bread slice inside it, resulting in the bread slice being placed on top of the toaster instead of inside it.

## B.3 Object states

We list the object state candidates considered for all objects that can change states in our experiments. In general, the states are assigned based on object properties.

| Object Type | Actionable Properties | Simulation States | Real-world States |
|---|---|---|---|
| Pot | Fillable, Dirtyable | filled, empty, dirty, clean | filled with water, empty |
| Faucet | Toggleable | toggled on, toggled off | toggled on, toggled off |
| StoveBurner | Toggleable | toggled on, toggled off | toggled on, toggled off |
| Bread | Sliceable | sliced, unsliced | N/A |
| Toaster | Toggleable | toggled on, toggled off | N/A |
| Fridge | Openable | open, closed | open, closed |
| Pan | Dirtyable | dirty, clean | N/A |
| Egg | Sliceable, Breakable | sliced, unsliced, cracked, uncracked | N/A |
| Potato | Sliceable, Cookable | sliced, unsliced, cooked, uncooked | N/A |
| Plate | Breakable (Some), Dirtyable | broken, dirty, clean | N/A |
| Microwave | Toggleable, Openable | toggled on, toggled off, open, closed | toggled on, toggled off, open, closed |
| Mug | Fillable, Breakable, Dirtyable | filled, empty, broken, dirty, clean | filled with coffee, empty |
| CoffeeMachine | Toggeable | toggled on, toggled off | toggled on, toggled off |
| HousePlant | Fillable | watered, not watered | N/A |
| Tomato | Sliceable | sliced, unsliced | N/A |
| Lettuce | Sliceable | sliced, unsliced | N/A |
| Bowl | Fillable, Breakable (Some), Dirtyable | filled, empty, broken, dirty, clean | N/A |
| Laptop | Openable, Toggleable, Breakable | open, closed, toggled on, toggled off, broken | N/A |
| Television | Toggleable, Breakable | toggled on, toggled off, broken | N/A |
| Carrot | Sliceable | N/A | sliced, unsliced |
| Drawer | Openable | N/A | open, closed |

Table 6: Object States

## B.4 Human Evaluation

Similar to the approach for human evaluation in Ahn et al. [7], we ask 2 groups of users, 3 in each group to compare the ground truth failure explanation labelled in the dataset and REFLECT-generated failure explanation for each failure scenario. The failure scenarios are randomly shuffled in the questionnaires sent to the users. The users are instructed to score 0 if the predicted explanation is incorrect, 1 if the predicted explanation is correct, and 2 if they are unsure. The final score reflected in the tables are the majority vote without counting "unsure". If there's a tie in the answers or more than one "unsure" is given, we will ask the users to re-score the specific case.

