# OpenReview forum: "REFLECT: Summarizing Robot Experiences for Failure Explanation and Correction"
_robot-learning.org/CoRL/2023/Conference — CoRL 2023 Poster_

### Official Review · Reviewer_GeFi · 2023-07-19

**Confidence:** 4
**Originality:** Good
**Technical Quality:** Very Good
**Clarity Of Presentation:** Very Good
**Impact:** 3

**Recommendation:**

Weak Accept: I recommend accepting the paper, but will not argue for my recommendation if the majority of other reviewers have a different opinion.

**Review:**

**Strengths**

1) The paper is well-written and easy to read.
2) The idea of converting multimodal sensory inputs into a unified summary and introducing a hierarchical structure in the summary is quite interesting.

 **Weakness**
1) The method used for extracting the visual and audio summary requires prior information (or a DSL) about the possible options for the spatial relations and the states of a given object. This kind of enumeration can be hard for a complex task.
2) The paper seems to assume information about the states that are possible for a given object. For example, the faucet can be turned on or off; the pot can be empty, filled and clean or filled and not clean.  However, it doesn't explain how this information about the possible states was obtained. The paper could possibly include a table in the appendix listing all the possible states for each object.
3) The heuristics used to determine the spatial relations may be good enough for the experiments made in the paper. However, they are brittle in general for objects with complex shapes. Such heuristics may provide a zero-shot generalization to novel scenes in a restricted environment, but I am concerned that the same may not hold true in a complex environment. Utilizing learning methods or pre-trained models to determine spatial relations appears to be a potential solution to address this concern.

**subtle issues**
1) The task-relevant scene graph is helpful for gathering the necessary information, allowing LLMs to focus on the relevant details when generating explanations and correcting the plan. However, what happens if the action failure affects other objects unrelated to the objects used in the plan? For example, a pot falling from the table may disconnect the power connection of the microwave. The robot needs to reconnect the power cable and turn on the microwave to cook a meal. However, there is no way the LLM knows if the power cable is connected or not; thus, getting a summary using task-relevant scene graphs cannot generate a feasible plan in such scenarios. In my opinion, predetermining a fixed task-specific subgraph may not be an ideal solution in complex scenarios.

**Quality Of The Limitations Section:**

Limitations are not well addressed

**Questions For Rebuttal:**

1) Please provide information about how the state of an object is determined. In particular, how do we know what text input needs to be given to the CLIP?
2) Please improve on the limitation section. The limitations sections mostly outline the future directions of this work. However, it does not describe much about the limitations of the work in consideration.

**Robotics Focus:**

Sufficient demonstration on hardware

**Summary Of Paper:**

This paper uses LLMs to detect, recover and explain errors that occur while executing the given plan. The primary objective is to convert the multimodal sensory data into a text summary that focuses on information about the objects relevant to the plan. Moreover, this approach introduces two additional abstractions over the summary derived from sensory inputs, refining it based on events and subgoal attainment. The subgoal-level summary serves as a means to query LLMs for error detection, while the event-based summary comes into play during the generation of explanations if any errors are detected. The failure explanation thus generated is used to correct the plan.

**Summary Of Recommendation:**

Overall, the paper is well written. The idea of getting a unified hierarchical summary is interesting but the techniques (in particular, heuristics) used to get the summary is brittle.

---

### Official Review · Reviewer_LGvx · 2023-07-20

**Confidence:** 3
**Originality:** Very Good
**Technical Quality:** Very Good
**Clarity Of Presentation:** Very Good
**Impact:** 4

**Recommendation:**

Strong Accept: I recommend accepting the paper and will argue for my recommendation even if other reviewers hold a different opinion.

**Review:**

**Strengths**

* The pipeline(s) for logging robot data and related events is impressive
* The prompt generation scheme used by the authors makes sense
* The ablation studies and the results adequately test the papers' claims on the systems contributed by the authors
* The authors leverage the LLM's ability to localize to the right level of explanation for the failure, which has been a big problem in the field of XAI. To the best of my knowledge, this is the first paper to do that directly from raw robot streams.
* The authors leverage the LLM to generate a high level plan for repair, which is impressive emergent behaviour
* The authors are contributing a failure dataset

**Scope for Improvement**

Among the limitations, the authors might do well to call out that their scenarios are all in static environments. Some aspects of their solution, particularly the scene graph generation, might be harder in dynamic environments.

**Quality Of The Limitations Section:**

Limitations are addressed clearly

**Questions For Rebuttal:**

None

**Robotics Focus:**

Sufficient demonstration on hardware

**Summary Of Paper:**

* The authors showcase REFLECT, a system for logging robot tasks, events, and sensory streams
* In order to log sensory streams, the authors use an ensemble of models to generate a scene graph. Changes in the scene graph are then logged as events (along with the end of robot actions). The logs are also annotated with subtask goals
* The scene graphs, the event annotations, and the subtask goals are then used to engineer prompts for an LLM to explain robot failures
* The authors show that with sufficient prompt engineering from the annotated data, the LLM is also able to construct a plan, which when grounded to skills, can often recover the error(s)
* Finally, and crucially for the community, the authors release a RoboFail dataset for the community to work on failure explanations and recovery. The dataset also includes an initial taxonomy of the failures.

**Summary Of Recommendation:**

I am strongly in favour of including this work in the technical program.

---

### Official Review · Reviewer_BfQU · 2023-07-20

**Confidence:** 5
**Originality:** Very Good
**Technical Quality:** Very Good
**Clarity Of Presentation:** Good
**Impact:** 3

**Recommendation:**

Weak Reject: I recommend rejecting the paper, but will not argue for my recommendation if the majority of other reviewers have a different opinion.

**Review:**

This paper has several strengths:

1. The problem of giving explanation to robot failures is useful and important - it is a crucial aspect of the feedback loop to both robot policy learning as well as LLM decision making.

2. One good insight of this paper is how more structured, task-oriented summary (captioning) of the scene is useful for robotics tasks, as opposed to using out-of-the-box image captioning models that are more general. This extracts information more informative to robot planning goals.

3. The creation of the robot failure dataset is a good contribution to the community in studying robot failures, and the taxonomy of failure classification is a good contribution as well.

Weaknesses:

1. The evaluation results involves some confusion; it seems that there are two levels of metrics that the paper wants to evaluate: 1) failure explanation correctness 2) how failure explanation contribute to task success. The 2) here is confusing as there have not been made clear whether it is the success of the planner, or the success of task execution. (See Questions for Rebuttal P4.) I think the paper's presentation of results can be made clearer so that the actual contribution is clear.

2. Related to Weakness P1 - I'm a bit uncertain about how far is the contribution of the paper, in the sense that there isn't too much evidence of how the failure explanation can contribute to better improvement of the policy performance. It is clear that this method presents a way to summarize robot failures, but it is unclear of how this could be used or its potential impact. I would be happy to hear the authors' thoughts on this.

3. In terms of the method, the scene graph of summarizing object relationships is useful, but it contains many hardcoded heuristics and seems a bit hacky, which cast doubts on the generalization of the method.

4. In terms of the experiments: it is not so clear about what specific setup of the robot policy is used here - it seems to be high-level LLM planning + some low-level controllers, but I don't really see how they interface with each other. It would be very helpful to explain what kind of policy this method applies to (and what the policy does not).

**Quality Of The Limitations Section:**

Additional details required

**Questions For Rebuttal:**

1. What is the low-level policy used for the real-robot experiments? Are they learned policy skills, or hardcoded pre-defined controllers? I did not find this information in the paper.

2. After passing in robot failures explanations to LLM, the LLM will come up with better planning and hence improve task performance; however, many failures (especially on real robot) will be due to low-level control failures. Can these failures be improved from robot explanation? It looks like there isn't really a module to improve low-level control policy.

3. For the failure correction planner: Is it replanning from the start, or continuing where the failure happened? If the robot do something irreversible and different from the previous plan (e.g. dropping a pan), can the planner come up with something corrective (e.g. picking up the pan) from there?

4. I have some confusion regarding Table 1: "Co-plan (correction planning success rate): percentage of tasks that succeed after correction"

     a) What does this metric mean? Do you actually execute the planned actions in the environment and check the actual policy performance, or check the correctness of the planner?

     b) The left table is "Execution Failure" - Why would the column "planning success rate" still apply to execution failure (what does this mean)? And why would generating failure explanation benefit the planning success of execution failure (since failure explanation is only used for the high-level planner in 3.3, I don't see how this affects the execution failure?)

5. The paper uses GPT4 as the LLM. How well do the other more accessible language models like GPT3.5 compare in terms of 1) original planning ability 2) correction after failure summarization? This is not really critical to the method itself, but curious how the method respond to different LLM - since GPT4 is not that accessible as of now and is very expensive, making reproducibility of the method hard. (again, it is not this paper's responsibility to show that all language models work well, but would be nice to give some knowledge of its sensibility. )

**Robotics Focus:**

Sufficient demonstration on hardware

**Summary Of Paper:**

This paper presents a method to summarize robot failure modes and use this information to improve LLM planning performance. It presents a novel method to create hierarchical summary of both planning and execution failures, and progressively generate failure explanation using LLM. It then uses the failure correction planner to generate a corrective plan to complete the task. The paper also contributes a robot failure dataset for future benchmarking.

**Summary Of Recommendation:**

I recommend Weak Reject since there are still some unaddressed questions in the paper; I would be happy to hear the author's thoughts and discuss more. It would be great if more information is given on 1) the scope of contribution of the paper 2) more details on the experiment setup 2) some clarifications of the questions above.

---

### Official Review · Reviewer_6L3t · 2023-07-20

**Confidence:** 3
**Originality:** Very Good
**Technical Quality:** Very Good
**Clarity Of Presentation:** Good
**Impact:** 3

**Recommendation:**

Weak Accept: I recommend accepting the paper, but will not argue for my recommendation if the majority of other reviewers have a different opinion.

**Review:**

This work is interesting as it further considers how LLMs can be used as planners, but in a way that takes more advantage of having a textual representation of the plan; the generated explanations would presumably be much appreciated by human bystanders or robot supervisors. There is reference to a human evaluation of the generated explanations that shows that they are indeed useful. The presentation is also thorough in its inclusion of quantitative and qualitative results for many reasonable ablations.

The primary weaknesses of this work come from missed connections with existing planning work.

The structure of the approach is parallel with existing planning systems while making no reference to this fact. Section 3.2 “Progressive Failure Explanation” is more commonly called “Execution Monitoring”. 3.3 “Failure Correction Planner” is called replanning. There is no other planner, it’s the same planner asked to make a new plan under new circumstances. It’s an interesting and quaint limitation of the LLM that it lacks the ability to plan unless it’s told explicitly how it messed up previously (as indicated by the w/o explanation ablation replanning success rate in Table 1.), a problem that most algorithms which people would call “planners” do not suffer from. For a representative historical example of these terms in the planning context, see 10.1109/70.370506. For more recent examples, simply search for "robot architecture" or "deliberative robot architecture."

Other downsides of completing basic reasoning tasks with tools that are not designed to complete reasoning tasks are not interrogated in this paper; LLMs are “a promising component for explainable and robust robotic systems” and the only “remaining challenge lies in how to generate a ‘textual summary’ of raw robot sensory data that can be used in prompts and a systematic way to query LLMs for accurate failure explanation”. Of particular interest to roboticists; how many queries to the model are made during a typical plan execution? At what computational cost and wall-clock compute time?

Short of crafting a baseline which more effectively couples the LLM with an existing planner (e.g. by producing PDDL from the summaries and invoking a planner), there needs to be more substantive discussion of the trade offs compared to using a conventional approach.

**Quality Of The Limitations Section:**

Additional details required

**Questions For Rebuttal:**

3.2 and Fig 3: Does subgoal verification happen after every subgoal during execution? It’s somewhat ambiguous, because it seems that it could also happen more frequently, as it would in a typical planning system. Otherwise, the plan will continue executing long past the moment of failure. However, if the system is prompting after every subgoal, during successful execution, then the plan analysis stage will still prompt for an explanation of what’s wrong with the plan. Would this cause problems?

As with conventional planning systems, a key challenge is digesting the world state into a symbolic representation and grounding a plan into actions on particular parts of the world state. This is aided by having relatively clean environments with few duplicated objects and no other activity occurring around the robot; when there are more objects in the scene, there are more chances for perception to fail, potentially causing the robot to act on the incorrect object, or to believe there is no object to act on. Unlike a conventional planning system, it isn’t immediately clear if the approach would still work well in the presence of more objects and other activity _even with perfect perception_; at what point (if any) does the size of the textual description of the scene cause the model to lose track? If additional activities around the robot create additional extraneous event-based summaries, at what point does the volume of these summaries become a problem?

The text of this paper has had its character spacing reduced making it difficult to read. Please fix your template.

**Robotics Focus:**

Sufficient demonstration on hardware

**Summary Of Paper:**

This paper describes an LLM-based plan execution monitoring approach, focusing on its ability to summarize and explain failures, and—because the LLM is also the planner— to generate new plans based on the failure explanation. Per-frame observations are formed by converting a scene graph representing objects and their spatial relationships into text and combining it with a caption of the audio observation and the current action in the plan. The LLM is given the subgoal observations and asked to verify that each subgoal has been achieved. If the subgoal hasn’t been achieved, the sequence of event-based observations is fed to the model and an explanation for a failure occurring within the observation window requested. Otherwise, an explanation for an issue with the plan is requested. These explanations serve as context for a request for a new plan. Experiments in a simulated environment and with a toy kitchen environment show that the approach outperforms several reasonable ablations.

**Summary Of Recommendation:**

This approach is interesting and makes improvements in a vein that many are exploring. It does not make sufficient connections to the existing methods it is recreating, nor critically evaluate the ways in which it does or does not surpass them. These issues are surmountable without major revision, but revision is necessary.

Update after rebuttal: the authors have adjusted the framing of the paper to better position the method and its strengths against possible alternatives. I think readers may still find the experiments confusing (i.e. including an evaluation of the method as a re-planning mechanism, where the value proposition is weaker), however there are likely many readers who will be interested in the overall scheme that the paper contributes, so I have raised my score to weak accept.

---

### Author Response · Authors · 2023-08-09
**Summary of the Rebuttal by Paper48 Authors**

First, we would like to thank all reviewers for their constructive comments and thorough reviews! We are happy that all reviewers find our work interesting and a good contribution to the community.

We have addressed each reviewer’s concerns individually. Following their suggestions, we have also made a few major changes to our paper:
* An updated introduction clarifying the scope and contribution of this work.
* More discussion on connections with existing planning works in the related work section.
* Clarifications on the experiment setup in the evaluation section.
* An updated limitation section discussing limitations of the scene graph generation method, the environment assumption of the framework, and the correction planner.
* An additional experiment comparing GPT-3.5 with GPT-4 on the RoboFail benchmark.

Thanks again to all reviewers for taking the time and efforts to help improve the paper.

---

### Author Response · Authors · 2023-08-15
**Followup**

Dear reviewers,

We'd like to reach out again to check if there are additional questions or concerns we can address before the rebuttal period ends today. Thanks again for taking the time to read our work and provide helpful feedback!

Paper48 authors

---

### Decision · Program_Chairs · 2023-08-30

**Decision:**

Accept (Poster)

**Comment:**

The paper proposes an approach to analyse plan errors during execution and provide explanations for why failures happen and how to correct them. The approach is situated as an offline method that analysis execution of a vision based task planner operating on a scene graph representation of the scene and uses pre-defined skills/policies that can be composed for a goal-directed plans. The approach enters around using LLMs for the failure explanation and correction task and tackles the issue of summarising experience (using multi-modal representations), how to generate task-relevant explanations using progressive interaction with an LLM and  how to ameliorate errors by replanning actions.

The esteemed reviewers have provided detailed feedback on the paper suggesting sharpening of the core contributions in relation to closely related approaches, better clarifications on the assumptions/limitations and more clarity on the evaluation metrics. The authors engaged with the reviewers and updated text in the manuscript.

Overall, the discussions lead to an observation that the positioning of the paper is somewhat in between two paradigms: is this a paper that explores if and to what extent LLMs can be used for error recovery planning from offline data or a paper that advances error recovery and re-planning by improving speed, robustness etc.  We would request the authors to deliberate carefully on this question and align the writing and experiments accordingly. If the paper is to be viewed as the former then the experiments with more LLMs would be appreciated guiding the community forward on how to use pre-trained LLMs for this (and closely related tasks). If the paper is positioned as the latter, then engagement with classic and contemporary work in plan recovery would be appreciated.